# Climate sensitivity estimates – sensitivity to radiative forcing time series and observational data

Ragnhild Bieltvedt Skeie[1], Terje Berntsen[1,2], Magne Aldrin[3], Marit Holden[3], and Gunnar Myhre[1]

[1]CICERO-Center for International Climate and Environmental Research – Oslo, PB. 1129 Blindern, 0318 Oslo, Norway.
[2]Department of Geosciences, University of Oslo, PB. 1047 Blindern, 0316 OSLO, Norway.
[3]Norwegian Computing Center, PB. 114 Blindern, 0314 Oslo, Norway

*Correspondence to*: Ragnhild Bieltvedt Skeie (r.b.skeie@cicero.oslo.no)

**Abstract.** Inferred Effective Climate Sensitivity (ECS$_{inf}$) is estimated using a method combining radiative forcing (RF) time series and several series of observed ocean heat content (OHC) and near-surface temperature change in a Bayesian framework using a simple energy balance model and a stochastic model. The model is updated compared to our previous analysis by using recent forcing estimates from IPCC, including OHC data for the deep ocean, and extending the time series to 2014. In our main analysis, the mean value of the estimated ECS$_{inf}$ is 2.0°C, with a median value of 1.9°C and a 90% credible interval (CI) of 1.2-3.1°C. The mean estimate has recently been shown to be consistent with the higher values for the equilibrium climate sensitivity estimated by climate models. The transient climate response (TCR) is estimated to have a mean value of 1.4°C (90% CI 0.9 - 2.0°C), and in our main analysis the posterior aerosol effective radiative forcing is similar to the range provided by the IPCC. We show a strong sensitivity of the estimated ECS$_{inf}$ to the choice of a-priori RF time series, excluding pre-1950 data and the treatment of OHC data. Sensitivity analysis performed by merging the upper (0-700m) and the deep ocean OHC or using only one OHC data set (instead of four in the main analysis), both give an enhancement of the mean ECS$_{inf}$ by about 50% from our best estimate.

## 1 Introduction

A key question in climate science is how the global mean surface temperature (GMST) responds to changes in greenhouse gases or other forcings. The climate sensitivity is determined by complex feedbacks that operate on very different timescales and may depend on the transient climate state. The standard metric for climate sensitivity is the equilibrium climate sensitivity (ECS) (or Charney sensitivity) given as the change in temperature at equilibrium for a doubling of $CO_2$, neglecting long-term feedbacks associated with the vegetation changes, carbon feedbacks and ice sheet dynamics. Estimates of the ECS are either based on complex climate models or observations of past climate (Collins et al., 2013;Knutti et al., 2017). The

Intergovernmental Panel on Climate Change (IPCC) presented a likely (>66% probability) range for ECS of 1.5 to 4.5°C (Collins et al., 2013).

Regarding the Earth as a climate laboratory and the changes in atmospheric composition and land use over the historical record as a perturbation experiment, observational based analysis of Earth's Energy Budget have been used to infer the climate sensitivity (Forster, 2016). Since the current climate is in a non-equilibrium state, observationally based methods can only account for the feedbacks operating during the historical period. These methods using the historical period with observations are referred to as inferred estimates (Armour, 2017;Forster, 2016) and have only the capability to derive an effective climate sensitivity and are generally significantly lower than ECS estimates from Atmosphere-Ocean General Circulation Models (AOGCMs) (Armour, 2017;Knutti et al., 2017).

Since IPCCs fifth assessment report (AR5) there has been an improved understanding of the causes of the differences in estimates of climate sensitivity from climate models and observational based methods, directed to two main reasons. First, recent analysis of time-varying feedbacks in AOGCMs simulations from Coupled Model Intercomparison Project Phase 5 (CMIP5) (Proistosescu and Huybers, 2017;Armour, 2017;Andrews et al., 2015) have indicated that in most AOGCMs the net feedbacks become more positive over time as a new equilibrium is approached. This is most likely due to evolution of the pattern of sea surface temperature increase in the Pacific and Southern Ocean and associated cloud feedbacks. Whether this slow warming has manifested itself in the climate record used for the analysis is the difference between effective and equilibrium climate sensitivity (Armour, 2017;Knutti et al., 2017). Second, ECS formally refers to global near-surface air temperature ('tas' in CMIP5 nomenclature) and in observational based methods observed surface temperature records that are a blend of air temperature over land and sea surface temperature (SST) over ocean are used in the estimation. Several observed surface temperature records exist with different methods to account for gap in the observations. Differences in historical surface temperature warming among various analysis is more than 0.1°C (Haustein et al., 2017) arising mainly due to approaches taken in regions missing or limited spatial coverage of observations. According to Richardson et al. (2016), there is a general bias in the surface temperature records since water heats slower than the air above and due to undersampling in fast warming regions (e.g. the Arctic). Taking both effects into account, Armour (2017) shows that previous estimates of $ECS_{inf}$ of about 2.0°C are consistent with estimates of ECS of 2.9°C from climate models.

Although it is now established that the ECS estimated by the use of complex climate models and $ECS_{inf}$ estimated by using historical observations would differ, there is still considerable spread in ECS estimates from models and between observationally based $ECS_{inf}$ estimates. The observational based methods and using complex models are complementary approaches to quantify the net effect of the feedbacks that determines the climate sensitivity. Complex climate models include processes that are highly parameterized, in particular the representation of clouds, precipitation and convection, and associated feedbacks, which are crucial for estimating the ECS (Bony et al., 2015;Tan et al., 2016). There are also a large spread in observational based estimates (Knutti et al., 2017). Better understanding of the feedbacks in the complex models as well as improvements and understanding differences among the observational based methods are needed.

Observational estimates of climate sensitivity can be improved using longer data series of higher quality (e.g. correcting for observational biases in temperatures or better forcing estimates) (Urban et al., 2014). Estimates can also be improved by including observational data on other climate variables, which were not previously available. Several studies indicate that the temporary slowdown in GMST in the beginning of the millennium coexisted with increased accumulation of heat in the deep ocean (e.g. Meehl et al., 2011;Meehl et al., 2013;Balmaseda et al., 2013;Watanabe et al., 2013;Chen and Tung, 2014;Lyman and Johnson, 2013). Johansson et al. (2015) found that if OHC change below 700m over this period were included in their observational based methods the mean value of ECS$_{inf}$ increased.

In this study we use our estimation model that were first documented in Aldrin et al. (2012) and further developed in Skeie et al. 2014. Our method is more complex than the common energy balance based estimates (Forster, 2016) in that we embed a simple climate model into a stochastic model with radiative forcing time series as input, treating the northern and southern hemisphere (NH and SH) separately and includes a vertical resolution of the ocean (40 layers). The radiative forcing time series are linked to the observations of OHC and temperature change through the simple climate model and the stochastic model, using a Bayesian approach. A unique feature with our method is that we use several observational datasets. The method estimates not only the ECS$_{inf}$ but simultaneously also provides posterior estimates of the radiative forcing, as well as posterior uncertainty estimates in the observations datasets and correlations between them. In this study we further develop our estimation model with additional observational datasets, including heating rates of the deep ocean (below 700m), new forcing time series from the IPCC AR5 as well as extended time series from 2010 to 2014 to update our estimate of ECS$_{inf}$. We carry out a number of sensitivity experiments to investigate causes of differences in observational based ECS$_{inf}$ estimates due to differences in the input data (observations of surface temperature, OHC and RF).

## 2 Data and methods

### 2.1 The model

Our full model consists of a simple climate model (SCM) with an idealized representation of the Earth's energy balance, a data model that describes how observations are related to the process states, and finally a parameter model that expresses our prior knowledge of the parameters (Aldrin et al., 2012).

The core of our model framework is the SCM, a deterministic energy balance/upwelling-diffusion model (Schlesinger et al., 1992). The SCM calculates annual hemispheric near-surface temperature change (blended SST and surface air temperature) and changes in global OHC as a function of estimated RF time series. The vertical resolution of the ocean is 40 layers down to 4000m. The output of the SCM can be written as $m_t(x_{1750:t}, \theta)$, where $x_{1750:t}$ (the RF from 1750 until year t) and $\theta$ are the true, but unknown, input values to the SCM. $\theta$ is a vector of seven parameters, each with a physical meaning. One of these parameters is the climate sensitivity, and the other parameters determine how the heat is mixed into the ocean, which includes the mixed layer depth, the air-sea heat exchange coefficient, the vertical diffusivity in the ocean and the upwelling velocity (see Schlesinger et al. (1992) and Aldrin et al. (2012) for details).

The true state of some central characteristics ($\boldsymbol{g_t}$) of the climate system in year t with corresponding observations can then be written as $\boldsymbol{g_t} = \boldsymbol{m_t}(\boldsymbol{x_{1750:t}}, \boldsymbol{\theta}) + \boldsymbol{n_t}$, where $\boldsymbol{n_t}$ is a stochastic process, with three terms, representing long-term and short-term internal variability and model error. For the short-term internal variability, we use the Southern Oscillation index (Table 1) to account for the effect of ENSO. The term for the long-term internal variability were implemented in Skeie et al. (2014) and the dependence structure of this term (i.e. correlations over time and between the three elements) is based on control simulations with a GCM from CMIP5 (see Skeie et al., 2014 for details) This term will also represent other slowly varying model errors due to potential limitations of the SCM and forcing time series. The third error term is included to account for more rapidly varying model errors.

For the (available) long-term observational data that defines $\boldsymbol{g_t}$ we consider the surface temperatures separately for the northern and southern hemispheres and the OHC separately for 0-700m and below 700m. Each of these elements of $\boldsymbol{g_t}$ are associated with one or more corresponding observational-based data series (Table 1), with individual error terms. To gain as much information as possible, we use several data sets for the same physical quantity (e.g. OHC above 700m) simultaneously (Aldrin et al., 2012;Skeie et al., 2014). Most of the data series are provided with corresponding yearly standard errors (Fig. S7a). However, we only use the temporal profiles of the reported errors and estimate their magnitudes within the model, taking into account the possibilities that the reported standard errors may under- or overestimate the true uncertainty (Appendix A and Aldrin et al., 2012; Skeie et al., 2014).

The unknown quantities are given prior distributions as presented in Skeie et al. (2014). The ECS is given a vague prior, uniform (0,20) and the informative priors for $\boldsymbol{\theta}$ based on expert judgment are listed in Table S1. We apply a Bayesian approach in the spirit of Kennedy and O'Hagan (2001) on calibration of computer models and use Markov Chain Monte Carlo (MCMC) techniques to sample from the posterior distribution (Aldrin et al., 2012).

## 2.2 Set up

The starting point, here called case A, is the main results from Skeie et al. (2014) (hereafter named Skeie14) with some modifications (see Appendix A). These modifications changed the mean $ECS_{inf}$ value from 1.8°C (median 1.7°C, 90% credible interval (CI) 0.92-3.2°C) to 2.0°C (median 1.8°C, 90% CI 1.0-3.4°C) (Fig. 1a, case A). The transient climate response (TCR) is calculated by running the model with 1% per year increase in $CO_2$ using the joint posterior distribution of the model parameters. These modifications increased the mean value of TCR from 1.4 to 1.5°C and the 90% CI shifted slightly to larger values (Fig. 1b).

In case A, we used four hemispheric pairs of observational based estimates of surface temperatures from about 1880 to 2010 and three series for OHC above 700m from about 1950 to 2010, and RF from Skeie et al. (2011,2014) (Table 1). The forcing time series used in case A are hereafter named Forc_Skeie2014 and the priors of each forcing mechanisms included (Table S2) are described in detail in the appendix D of Skeie14.

The potential for improving the constraint of the estimate of the climate sensitivity using observationally based methods, depends crucially on the quality of the input forcing data and the quality and amount of observational data. In case B, we

include new and improved knowledge of the forcing time series and add new data for OHC below 700m and observational data are extended to 2014. More specific in case B we 1) replaced the Forc_Skeie14 prior with the AR5 effective radiative forcing (ERF) estimates (Myhre et al., 2013) hereafter named Forc_AR5. The priors for the forcing mechanisms included (Table S2) are constructed to be consistent with the uncertainties provided in AR5 and the same relative uncertainty for the prior forcing is used over the entire time period. ERF includes rapid adjustments allowing the full influence on clouds except through surface temperature changes (Sherwood et al., 2014;Boucher et al., 2013;Myhre et al., 2013). 2) Include data for OHC below 700m (ORAS4) and add one extra data series for OHC above 700m (also ORAS4). Note that the deep ocean OHC is added as a separate dataset and not merged with the upper ocean. Including data on OHC in the deep ocean thus has the potential to better constrain the parameters in the SCM that determine how the heat is mixed into the ocean as well as the posterior estimates of the effective radiative forcing. 3) Use updated versions of the data prior to 2010, and 4) extend the time series from 2010 to 2014.

Previous studies using similar methods have obtained different results with respect to the estimated $ECS_{inf}$ (Knutti et al., 2017). We perform three sensitivity experiments to investigate the effects of different choices about how to use OHC data (cases C and D, sect. 4.1) and how sensitive the results are to pre-1950 data (case E, sect. 4.2).

## 3 Improved estimate of inferred effective climate sensitivity

Here we present our revised estimate of $ECS_{inf}$ by replacing the RF prior with IPCC data, including OHC data below 700m and extending the time series to 2014 (Case B). We consider this analysis using the IPCC forcing estimates, including deep ocean OHC and extending the length of the input data series as the most trustworthy and physical based case and thus regard it as our main estimate of the $ECS_{inf}$, with a mean of 2.0°C (median 1.9°C, 90% CI 1.2-3.1°C). The mean value is similar while the 90% CI is narrower compared to the refined Skeie14 estimate (Fig. 1a). The individual influence of the four major updates between case A and B is shown in Fig. S1 and described at the end of this section. The mean value of TCR in case B is 1.4°C (median 1.3°C, 90% CI 0.9-2.0°C) (Fig. 1b). As for the $ECS_{inf}$ estimate, the TCR mean value is similar and the 90% CI is narrower compared to the refined Skeie14 estimate (Fig. 1b). The GMST change is well reproduced (Fig. 2, case B), and less of the recent GMST change is attributed to long term internal variability compared to the refined Skeie14 estimate (Fig. S5a-b).

The rate of change in anthropogenic forcing is larger between 1940 and 1970 using Forc_AR5 compared to Forc_Skeie14 (Fig. 3). The fit to the GMST in the 1980s-1990s improved (Fig. 2 case B vs. A), where the root mean square error between 1980 and 1999 decreased from 0.12 to 0.077°C. Figure S5 shows posterior estimates of the long-term internal variability, the ENSO term and the model errors. Parts of the increase in GMST over the last decades are explained as long-term internal variability, but the amplitude decreases in case B compared to case A (Fig. S5a-b). In case B, the estimated amplitude of the multi-decadal internal variability (about 0.2°C in each hemisphere, cf. Figure S5) is in good agreement with the decadal trends

in global surface temperatures found in unforced control simulations in the multi-model ensemble from CMIP5 (0.2-0.4°C, Palmer and McNeall, 2014).

The prior anthropogenic mean forcing in 2010 increased from 1.5 to 2.3 Wm$^{-2}$ from case A to case B when Forc_AR5 replaced Forc_Skeie14. For case A, the posterior forcing is shifted to higher values compared to the prior, suggesting that the historical
data and our method supports higher forcing than the Forc_Skeie14 prior. When the prior is changed to Forc_AR5 in case B, the posterior for the anthropogenic forcing is much closer to the prior (Fig. 3), which indicates that the method and observational data is more in accordance with the new prior than the old one. The same holds for the total forcing (Fig. S4). The 90% CI for the posterior anthropogenic forcing was 1.3 to 2.8 Wm$^{-2}$ in case A compared to 1.3 to 3.4 Wm$^{-2}$ in case B. The upper limit of the 90% CI is shifted to larger values. The most uncertain part of the forcing time series is associated with
aerosols. The difference between the two forcing priors is mainly due to a much weaker aerosol forcing in Forc_AR5 than in Forc_Skeie14 (compare the two dashed-dotted error bars in Fig. 4a). While the posterior aerosol forcing where shifted to smaller negative values in case A, the prior and posterior for aerosol forcing is similar in case B (Fig. 4b). A relative weak aerosol-cloud interaction as included in Forc_AR5 is consistent with the recent findings in Malavelle et al. (2017) on how sulphate aerosols from volcanic emissions influences clouds.

The ERFs in AR5 are based on an assessment of several studies reflecting improved knowledge of the forcing mechanisms compared to the one-model RF results used in Skeie14. The new ERFs gave a better posterior estimate of GMST (Fig. 2) and reduced change from prior to posterior forcing (Fig. 3). Remark that the number of forcing time series that can be combined was 18 in Skeie14, including three timeseries for volcanic and eight for aerosols, compared to only one time series for each of these forcing mechanisms in Forc_AR5 (Table S2). This gives less flexibility in the time development of the forcing in case
B compared to case A, however the GMST change is better reproduced in the 1980s-1990s using Forc_AR5 compared to Forc_Skeie14.

Ultimately, global climate change is governed by the radiative imbalance at the top of the atmosphere (TOA) and modulated by the internal variability. Forcing by greenhouse gases and aerosols as well as albedo changes, feedback processes and the radiative responses to temperature changes determine this imbalance. With a positive net imbalance at TOA, energy
accumulates in the Earth system, mainly as increasing OHC (Church et al., 2011). Since OHC is the dominant energy storage in the system, these data series have profound influence on the ECS$_{inf}$ estimates (Tomassini et al., 2007;Skeie et al., 2014;Aldrin et al., 2012;Johansson et al., 2015). In case B, we have extended our use of OHC data, so in addition to the three OHC data series above 700m we have included the ORAS4 data above and below 700m (Table 1) as two separate data sources. Including the deep ocean OHC data gives a stronger constraint on the overall accumulation of heat in the system, and the posterior
estimates of the parameters of $\boldsymbol{\theta}$ that determine the vertical transport of heat in the ocean, the effective diffusivity and the upwelling velocity increase by 44 and 31%, respectively. Having separate data series for the two ocean layers also provides information that influences the balance between negative (by aerosols) and positive forcings, since these forcings have different evolution over time (cf. sect. 4.1).

In Fig. 5 the observed and fitted OHC for case A and B are shown. Including data on OHC change below 700m increases the total heat uptake. The increase in the fitted OHC above 700m over the last decade is larger in case B compared to case A. In case B the increase in the fitted OHC above 700m is larger than the observational data, while below 700m, the observed OHC increase is higher than the fitted one (Fig.5). This is to be expected since the parameters of $\boldsymbol{\theta}$ do not change over time. Thus, the observed rapid change in OHC below 700m over the last years with corresponding slower warming above 700m, is attributed to long-term internal variability (a part of the $\boldsymbol{n_t}$ term) in the model (Figure S5c-d). Remark that the Ishii and Kimito series is out of the 90% CI. The reason is that the assumed observational errors for all series are much larger back in time than in the recent years (see Appendix A). Therefore, the various data series are aligned quite close to each other in the recent years, and since the Ishii and Kimoto series has a much weaker trend than the others, it lies above the 90% CI in the first part of the data history.

The update of the $ECS_{inf}$ from case A to B was done stepwise in four steps (Fig. S1f, g, i and j). The new ERFs were first implemented. The posterior forcing is much closer to the prior using Forc_AR5 instead of Forc_Skeie14, and also the fit to the GMST in the 1980s-1990s improved with a decrease in the root mean square error between 1980 and 1999 from 0.12 to 0.087ºC compared to case A. The stronger forcing resulted in a shift of the $ECS_{inf}$ estimate to lower values (Fig. S1f vs. e), with an $ECS_{inf}$ mean value of 1.5ºC (90% CI 0.9-2.3ºC). So far, only OHC data in the upper 700m were used, leaving the model unconstrained with respect to the heating of the deeper ocean.

We then included the ORAS4 data above and below 700m as two separate data sources. Similar to Johansson et al. (2015) we found that including the OHC change below 700m increases the total heat uptake and thus the mean value of $ECS_{inf}$ from 1.5 to 1.7°C (Fig. S1g vs. f). The 90% CI shifted to larger values ranging from 1.0-2.8°C.

The last two steps to update the $ECS_{inf}$ estimate from case A to case B was to use the most recent version of the data prior to 2010 and to extend the data series used from 2010 to 2014 (Table 1). Some of the observational data series have been updated by the data suppliers, so first we use refined data up to 2010 before we extend the data series to 2014 (cf. Appendix B). Using the refined data up to 2010, the estimated mean $ECS_{inf}$ increased from 1.7 to 2.0°C (Fig. S1i) and the 90% CI was shifted again to larger values ranging from 1.1-3.3°C. Further, when the data series were extended from 2010 to 2014 the upper bound of the 90% CI decreased from 3.3 to 3.1°C while the lower bound remained unchanged and the mean estimate slightly reduced (Fig. S1j).

In total, the changing from case A to Case B did not change the mean value of $ECS_{inf}$ (it is 2.0ºC in both cases), but the 90% CI was reduced from 1.0-3.4 ºC to 1.2-3.1ºC. The reduction in $ECS_{inf}$ in the first step of the update is more or less counteracted by the subsequent steps.

## 4 Sensitivity tests – the use of input data

We now investigate possible causes of differences in observational based $ECS_{inf}$ estimates due to the use of input data. We analyze the impacts of different usage of the OHC data (cases C and D) and the treatment of uncertainties in the GMST data (case E).

### 4.1 The role of the use of OHC data

The vertical transport of heat in the SCM (with 40 vertical layers) is quite simple. Turbulent diffusion mixes heat down from the surface, while downwelling transports heat directly to the deepest layer, i.e. no detrainment to intermediate layers (Aldrin et al., 2012). Therefore, it is of interest to investigate whether a constrain of the model with OHC data for the total depth of the ocean instead of above and below 700 m. In Case C we do not separate the 0-700m from the deeper ocean. We use four data sets for total OHC by adding the ORAS4 below 700m data to each of the four OHC above 700m estimates. Merging the OHC above and below 700m (Case C) results in a substantial decrease in the posterior ERF from 2.5 to 1.8 Wm$^{-2}$ (Fig. S6b-c) and an increase in the $ECS_{inf}$ estimate from a mean value of 2.0°C (median 1.9°C) to 3.2°C (median 2.9°C) (Fig. 1a). Without the separate constraint on the OHC above and below 700m, the posterior warming of the ocean increases faster (compared to case B) over the last 20 years (Fig. 6). This is mainly caused by enhanced warming in the upper 700m (Fig. 7). This allows for a stronger negative ERF estimate for aerosols (Fig. 4a). While the prior and posterior radiative forcing in Case B is similar, in case C the posterior aerosol ERF is shifted to lower values (Fig. 4a), the posterior net forcing is shifted towards lower values (Fig. 4a and Fig. S6c) and hence a higher estimated $ECS_{inf}$ (Fig. 1) compared to case B. This anti-correlation between aerosol forcing and $ECS_{inf}$ is illustrated in Fig. 4c for case B. However, the observations show a stronger recent increase in heat in the deep ocean (c.f. sect. 3) and not in the upper 700m, so this test where this information is not used is likely to overestimate the aerosol forcing strength and hence overestimate the $ECS_{inf}$. Since the IPCC best estimate of -0.9 Wm$^{-2}$ was published in 2013 for aerosols ERF, studies point towards weak aerosol-cloud interaction (Gordon et al., 2016;Malavelle et al., 2017;Toll et al., 2017). These recent studies indicate that there is no firm evidence to revise the IPCC AR5 aerosol ERF best estimate yet. We therefore keep case B as our best estimate, since having separate data series for the two ocean layers provides information that constrain the balance between negative and positive forcings, due to their different time evolution.

A unique feature with our method is that we use data from more than one observational dataset. It is obvious that, as long as the various data series for the same quantity (here OHC above 700m) differ, it is easier to fit a model to one data series, thus giving less uncertainty in the posterior estimates. In case D we test the effect of using one alternative time series for OHC. We choose to use the Levitus2000 time series, that is the same OHC data as used in Johansson et al. (2015). The pentadal heat content are used from 1955 to 2012, treated as annual observations, and extended to 2014 using the yearly OHC data for the upper 2000m from the same data source. We use the OHC data for the upper 2000m as they were data for the total OHC. Observed energy stored below 2000m is not included in the estimation and hence the ECS might be underestimated. Energy stored below 2000m is uncertain. Purkey and Johnson (2010) found an increase in OHC in the abyssal and deep Southern

Ocean in the 1990s and 2000s based on sparse observations from ships, but it is not clear if it is a long-term trend. Llovel et al. (2014) could not detect deep-ocean (below 2000 meter) contribution to sea level rise and energy budget between 2005 and 2013 using ocean observations and satellite measurements, however the uncertainties are large.

As in case C, we do not separate the OHC data above and below 700m. Quite similar to case C, there is a more rapid increase in the posterior estimate of total OHC (Fig. 6) compared to case B, the increased warming is mostly in the upper 700m (Fig. 7) and the posterior forcing is shifted to lower values than in the prior (Fig. 4a and Fig. S6d). In case D the estimated mean $ECS_{inf}$ is 2.8°C (median 2.6°C, 90% CI 1.5-4.6°C) (Fig 1a, case F). This is higher than in case B, but lower than for case C. The estimated total OHC has a narrower range when OHC above and below 700m are merged (Fig. 6, left panel). The range is also narrower in case D than in case C. As expected, using several data series for OHC (Case B: 5, Case C: 4, Case D: 1) increase the posterior observational error. Note that the magnitude of the observational errors are estimated (Aldrin et al., 2012;Skeie et al., 2014). In case D, the posterior standard deviation of the observed OHC is similar to the reported standard deviation (Fig. S8), while using several OHC time series the posterior standard deviation is larger (Fig. S7) and arguably more correct than reported due to the large variability among the datasets (Appendix A). Hence, larger uncertainties in the observed OHC data result in larger uncertainties in the estimated OHC.

Johansson et al. (2015) used the same OHC data series as in our case D and a similar method, however their 90% CI for the OHC in the upper 2000m (their Fig. S5) is even narrower. This might not only be due to the use of one OHC dataset. While we estimate the magnitude of the observational error, Johansson et al. (2015) use the error given by the OHC data provider. In Johansson et al. (2015) the estimated uncertainties in OHC were smaller than the given observational uncertainties (their Fig. S5). The narrower $ECS_{inf}$ range may primarily be because Johansson et al. (2015) assumed very small measurement errors in the most informative data (OHC), secondly that they ignored time correlation in observational errors and did not take into account long-term internal variability in the same degree as in our method.

To sum up, using several observational series (and estimate observational errors) increase the estimated observational errors to more realistic values, since data series are not well correlated, and hence increase the range of estimated OHC with implications on estimated $ECS_{inf}$.

**4.2 The role of uncertainty estimates in the temperature series**

The prior standard deviation for the surface temperature data are quite different among the data sets (Fig. S7a). The NCDC data has 3 to 5 times larger standard error prior to 1950 compared to after 1950, while it is more constant back to the 19[th] century for the three other data sets.

To investigate this, we re-estimated our model using data only after 1950, which is equivalent to assuming a very large uncertainty prior to 1950. The estimated magnitude of the ENSO signal increases (Fig. S5a-b) since the data series are more correlated in the latter part of 20[th] century. For temperature, the model fits well to the observations of GMST, but with a larger 90% CI range (Fig. 2) and the observed NH and SH temperatures are well within the 90% CI of the model (Fig. S9). The mean $ECS_{inf}$ increases from 2.0 (median 1.9 °C) to 2.2°C (median 2.1°C) and the upper 90% CI limit increases from 3.1 to 3.8°C

(Fig. 1a, case E vs. B). The mean TCR increases from 1.4 to 1.5ºC and the 90% CI is shifted slightly to lower values compared to the range from IPCC by 0.1ºC (Fig. 1b).

Johansson et al. (2015) used only the NCDC data for GMST, thus the data prior to 1950 was given little weight when fitting the model. Our ENSO signal is now (case E) of similar magnitude as in Johansson et al. (2015) (their Fig. 1b). The $ECS_{inf}$ uncertainty in this study is still larger and our mean value is slightly higher than their lower limit of 2ºC.

Excluding data before 1950 also excludes the late 19[th] century period with a large volcanic eruption where the signal in the GMST data is small and quite uncertain (Santer et al., 2016). Santer et al. (2016) argued that the method in Johansson et al. (2015) down weights the volcanic forcing due to the small response of the Krakatau eruption in the temperature data. Johansson et al. (2016) responded that the observational uncertainty was large so the GMST data at that time will have a limited effect. In our results, excluding observations before 1950, the GMST response following the Pinatubo eruption in 1991 increases (Fig. 2) and are similar to observations due to the larger ENSO signal and stronger posterior volcanic signal.

In the early period, the aerosol forcing had a larger relative contribution to total ERF causing a more uncertain forcing trend in the early period. Uncertainty in the temporal trend of the forcing is not included, and better representation of forcing uncertainties than the scaling approach is needed (Tanaka et al., 2009). Omitting data before 1950 (case E), when the net forcing is more uncertain (Stevens, 2013), makes it easier to fit the model to observations but the uncertainty in estimated $ECS_{inf}$, TCR and GMST and increases (Fig. 1 and 2).

## 5. Discussions and conclusions

Causes of differences in observational based estimates of $ECS_{inf}$ due to the use of input data are analyzed and an updated $ECS_{inf}$ estimate is presented using our Bayesian estimation model. Adding observational data from 2011 to 2014, OHC data below 700m and replacing forcing data with IPCC AR5 ERFs, the $ECS_{inf}$ posterior mean was 2.0°C (median 1.9°C, 90% CI 1.2-3.1°C). The mean value is similar and the range is slightly narrower than the refined Skeie14 estimated (Fig. 1 case B vs. A). The mean $ECS_{inf}$ estimate is larger than in Skeie14. Although the estimate in case A and B are quite similar, the $ECS_{inf}$ estimate shifted to lower values when Forc_AR5 replaced Forc_Skeie14 from a mean $ECS_{inf}$ estimate of 2.0°C to 1.5°C and shifted to larger values when OHC data below 700m were included to a mean $ECS_{inf}$ value of 1.7°C. The $ECS_{inf}$ estimate was very sensitive to the forcing data used and we showed that the $ECS_{inf}$ estimate was also sensitive to the assumed uncertainties in the GMST data (Case E, $ECS_{inf}$ mean value increased from 2.0 to 2.2°C) and how the OHC data were treated (Case C and D, with mean $ECS_{inf}$ of 3.2 and 2.8°C respectively).

Bayesian methods have recently been reviewed by Annan (2015) and Bodman and Jones (2016) and limitation by assuming constant sensitivity over time, the role of the $ECS_{inf}$ prior distribution and equal efficacy for different forcings have been discussed. Implementing an alternative prior for $ECS_{inf}$ as in Skeie14, where $1/ECS_{inf}$ is uniformly distributed, shifted the mean $ECS_{inf}$ to lower values from 2.0°C (median 1.9°C, 90% CI 1.2-3.1°C) to 1.6°C (median 1.6°C, 90% CI 0.97-2.5°C). The $ECS_{inf}$ estimate is sensitive to the prior, however one could argue against this alternative prior because it has high probability for low

climate sensitivities that may not be realistic, with 76% probability for $ECS_{inf}$ being lower than the pure black-body radiation sensitivity of 1.1°C (Aldrin et al. 2012, Skeie et al. 2014). Recently, studies have suggested that assuming equal efficacy for all forcings bias the ECS estimate low (Marvel et al., 2015;Shindell et al., 2015) even when ERFs are used. In our approach, the efficacy is implicitly included in the forcing uncertainty and thus accounted for. However, if we apply an efficacy of 1.5

for ozone, surface albedo, BC on snow and aerosols, which is the efficacy found in the analysis of Shindell (2014), the probability density function of the ECS is shifted to larger values (Fig. S1l), with a 90% CI ranging from 1.2 to 3.7°C.

The fit to the temperature data in the 1980s and 1990s improved using Forc_AR5 instead of Forc_Skeie14 indicating that the forcing trend over this period is better represented in Forc_AR5 compared to Forc_Skeie14. The trend in the forcing is more uncertain in the first half of the 20th century due to less dominance of $CO_2$, and in our method the same relative uncertainty

for the prior forcing is used over the entire time period. A sensitivity simulation omitting observations before 1950, similar to making these observations very uncertain, gave better representation of the GMST in the latter part of the 20th century and an increased mean $ECS_{inf}$. Future work should include uncertainties in the temporal development of the forcing, and there is a clear need for an international effort to establish forcing time series, using a consistent forcing definition and allowing for uncertainties in emissions, to give a better representation of the temporal uncertainties.

Including OHC-data below 700m shifted the $ECS_{inf}$ to higher values. The estimated $ECS_{inf}$ was found to be very sensitive to how the OHC data were used. Including four OHC time series, but merging the data above and below 700m (case C), the $ECS_{inf}$ mean value increased from 2.0 to 3.2°C. The probability of $ECS_{inf}$ above 4.5°C increased to 13%, values that are practically excluded in our main estimate (case B). Previous studies have used total column OHC data and due to the simple representation of the ocean one can argue that this might be more appropriate. However, in case C most of the recent increase

in OHC in the model occurred in the uppermost 700m allowing a stronger aerosol cooling (Fig. 4a) and hence a larger $ECS_{inf}$, while the observations indicate that the ocean was warming mainly below 700m. Using only the total column OHC might therefore overestimate the aerosol forcing strength and hence the $ECS_{inf}$. We recognize structural uncertainties in the model, and a multi-model intercomparison of observational methods using identical input data would be of great value to investigate these uncertainties.

Using only the Levitus2000 series for OHC for the total ocean column (case D), the $ECS_{inf}$ 90% CI was shifted to lower values with a range from 1.5-4.6°C and the range shrunk compared to case C. The historical measurements of ocean temperatures are sparse (Abraham et al., 2013), with large differences between the datasets. The temporal structure of the reported uncertainties differs, and the full uncertainties are often not assessed. Hence, relying on only one OHC series and its reported uncertainty may underestimate the observational uncertainties and hence overestimate the certainties in the estimated OHC with

implications for the $ECS_{inf}$ estimate.

Recent studies indicate that the upper-ocean warming is underestimated due to the gap-filling methods (Durack et al., 2014;Li-Jing et al., 2015), in which case also the $ECS_{inf}$ will be underestimated. Refining historical OHC estimates, not only the best value, but also the uncertainty is crucial for observational based $ECS_{inf}$ estimation.

Other priorities are to improve the GMST series, including uncertainties, not only the recent trend (Karl et al., 2015;Cowtan and Way, 2014) but also for earlier time periods. Assuming a very large uncertainty prior to 1950, the GMST fit improved, ECS$_{inf}$ mean increased while the estimated uncertainty ranges increased.

Our ECS$_{inf}$ posterior mean was 2.0°C with 90 % CI of 1.2 to 3.1°C. This is consistent with a mean ECS of 2.9°C (Armour, 2017), which compares reasonably well with climate model estimates (Andrews et al., 2012;Forster et al., 2013). A final remark is that it is not obvious that the true ECS is a more relevant metric for the climate sensitivity than the ECS$_{inf}$ in a policy context (i.e. the Paris agreement). The United Nations Framework Convention on Climate Change (UNFCCC) has not adopted a pre-defined definition of GMST and the stronger long-term feedbacks found in analysis of CMIP5 simulations  (Proistosescu and Huybers, 2017) operates on a time scale longer than the timescale for reaching  2°C.

## Appendix A: Refinement of Skeie14

A few updates/corrections to Skeie14 (Fig. S1a) had to be made prior to the analyses presented in this study. In the Skeie14 study, the standard error of observed OHC above 700m for two out of the three series were constant in time, while for the third dataset the standard error decreased with time. Due to the limited observational data back in history (e.g.  Abraham et al., 2013), it is reasonable to assume that the shape of the standard error of observed global OHC increase back in time, as for the CSIRO series. Therefore, we now assume a common observational uncertainty temporal profile for OHC above 700m equal to CSIRO for all the OHC time series (Fig. S1b). Note that the magnitude of the observational errors are estimated in our approach (Aldrin et al., 2012;Skeie et al., 2014), i.e. we account for the possibilities that the reported observational errors may be biased upward or downwards compared to the real observational errors.

In fact, the results from Skeie et al. (2014, appendix B) indicated that the reported standard errors for the Levitus and the Ishii and Kimoto OHC series were too low. We have investigated this further by the following simple analysis:

Let $y_{1t}$ and $y_{2t}$ be two different estimates of the true OHC in year $t$. Then $y_{1t} =$ "$true\ OHC$" $+\ e_{1t}$ and $y_{2t} =$ "$true\ OHC$" $+\ e_{2t}$ . Here, $e_{1t}$ and $e_{2t}$ are error terms, with reported standard deviations $s_{1t}$ and $s_{2t}$ , and with true, but unknown standard deviations $\sigma_{1t}$ and $\sigma_{2t}$. The difference of the series is $y_{1t} - y_{2t} = e_{1t} - e_{2t}$, so even if we cannot observe the errors, we can observe their difference. If the two data series are based on more or less the same data, as for the OHC series used here, one can expect that $e_{1t}$ and $e_{2t}$ are positively correlated. Then $Var(y_{1t} - y_{2t}) = Var(e_{1t} - e_{2t}) <= (\sigma_{1t}^2 + \sigma_{2t}^2)$.

We can estimate the average variance of the differences $y_{1t} - y_{2t}$ over all time points by $Var^{obs} = 1/(n - 1) \sum_t (y_{1t} - y_{2t} - m)^2$, where $m$ is the average of $y_{1t} - y_{2t}$ and $n$ is the number of years. This could be compared to the corresponding reported variance under the assumption of uncorrelated errors, by $Var^{rep} = 1/n \sum_t (s_{1t}^2 + s_{2t}^2)$, and if the reported standard deviations are correct, then the variance ratio $Var^{obs}/Var^{rep}$ should be less than or equal to 1. For differences of the Levitus, Ishii and Kimoto and ORAS4 (above 700m) series, the variance ratios are between 2.13 and 3.74

(Table A1), indicating that the reported observational errors for these series are too low, and the real uncertainty may be larger. This is an additional argument for using the CSIRO standard errors for all OHC series.

Another update of Skeie14 that was needed, was to use monthly volcanic RF data (Fig. S1c) compared to yearly data in Skeie14. In addition to the three global mean surface temperature (GMST) time series used in Skeie14, another time series for
GMST has been published recently (Cowtan and Way, 2014). This time series find a stronger increasing trend in temperature over the last decade compared to the HadCRUT4 data, due to their method of accounting for the unsampled regions in the world. This data series is now included (Fig. S1d).

Our previous studies showed that the correlation between the observational errors in temperature data was almost uncorrelated with the observational errors in the OHC data. Therefore, to simplify the numerical computations, we from now on assume
that these correlations are exactly zero (Fig. S1e).

The estimated $ECS_{inf}$ for each step in the refinement of Skeie14 is presented in Fig. S1a-e.

## Appendix B: Extending data up to and including 2014

When extending the analysis from 2010 to 2014, not all the time series used in the estimation is available up to and including year 2014. Below is a description of how the different datasets are extended if not available up to 2014.

AR5 ERF: The end year for the forcing time series presented in AR5 is 2011 and has to be extended to 2014. For long-lived greenhouse gases the time series are extended using recent observations of global mean concentrations and the formulas relating concentrations and forcing used in Skeie et al. (2011). Tropospheric ozone, stratospheric ozone, aerosol ERF, land use change, BC on snow and volcanoes are kept constant 2011-2014. Stratospheric water vapor follow methane RF. Contrails RF is extended using aircraft traffic data (http://airlines.org/dataset/world-airlines-traffic-and-capacity/). Solar RF is extended
using the Physikalisch-Meteorologisches Observatorium Davos (PMOD) composite (Frohlich and Lean, 2004).

CSIRO: Data up to and including 2012 were downloaded. The time series were extended from 2012 to 2014 using the mean rate of change of the other OHC data. The uncertainty in 2014 and 2013 is set equal to the uncertainty in 2012.

ORAS4: Balmaseda et al. (2013) investigated the time evolution of global OHC at different depths of the ocean from 1958 to 2009 using the European Centre for Medium-Range Weather Forecasts ocean reanalysis system 4 (ORAS4). Five ensemble
members of ORAS4 are generated that sample plausible uncertainties in the wind forcing, observation coverage, and the deep ocean. The ORAS4 system runs automatically in operations, with numerical weather prediction forcing and observations that are not manually quality controlled. The 1x1-degree Ocean potential temperature up to December 2014 are made available through the APDRC (http://apdrc.soest.hawaii.edu/datadoc/ecmwf_oras4.php) for one ensemble member. The trend in OHC for the total depth and upper 700m from 2010 to 2014 based on the one ensemble member is used to extend the corresponding
OHC data for all the five ensemble members from Balmaseda et al. (2013) up to 2014. The data after 2009 are based on the automatic ORAS4 system, and not quality controlled and the results in this paper using the data after 2009 should be interpreted by caution. The same method is used to extend the ORAS4 data from 2009 to 2010 (Fig. S1g-i). From the five ensemble members the estimate with uncertainty is calculated as the annual average and standard deviation of OHC above and below

700m. The standard deviations are modified by smoothing the curve (9-year moving average) since the curve was otherwise very static.

**Data availability.** Several publicly available data sets were used in this study. The specific references to the data sources are given in Table 1. Model outputs are available upon request.

**Author contributions.** All authors designed the study and discussed the results; RBS and MH prepared the data; MH performed the simulations; RBS and MH made the figures; RBS prepared the paper with contributions from all co-authors.

**Competing interests.** The authors declare that they have no conflict of interest.

**Acknowledgements.** This research was partly supported by the Norwegian Research Council under the project EVA - Earth system modeling of climate Variations in the Anthropocene, grant number 229771. We kindly acknowledge the data providers listed in Table 1 for providing the data for the analysis.

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

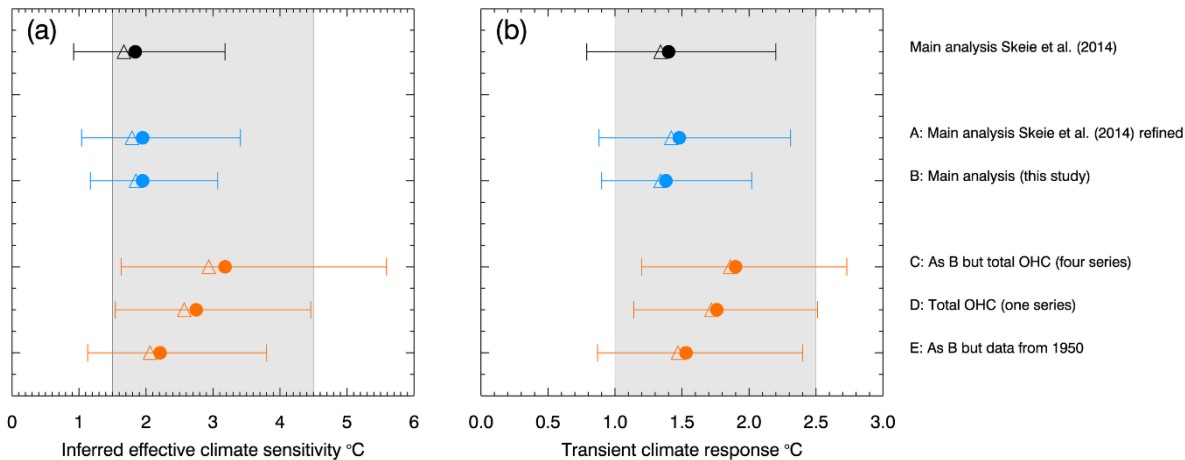

**Figure 1: Posterior 90% CI for ECS$_{inf}$ (a) and TCR (b) for the different analyses in this study. The estimated posterior mean is indicated by a dot and the median by an open triangle. The IPCC AR5 likely range (>66% probability) for ECS (a) and TCR (b) is presented as gray shadings. Fig. S2 show the corresponding probability density functions.**

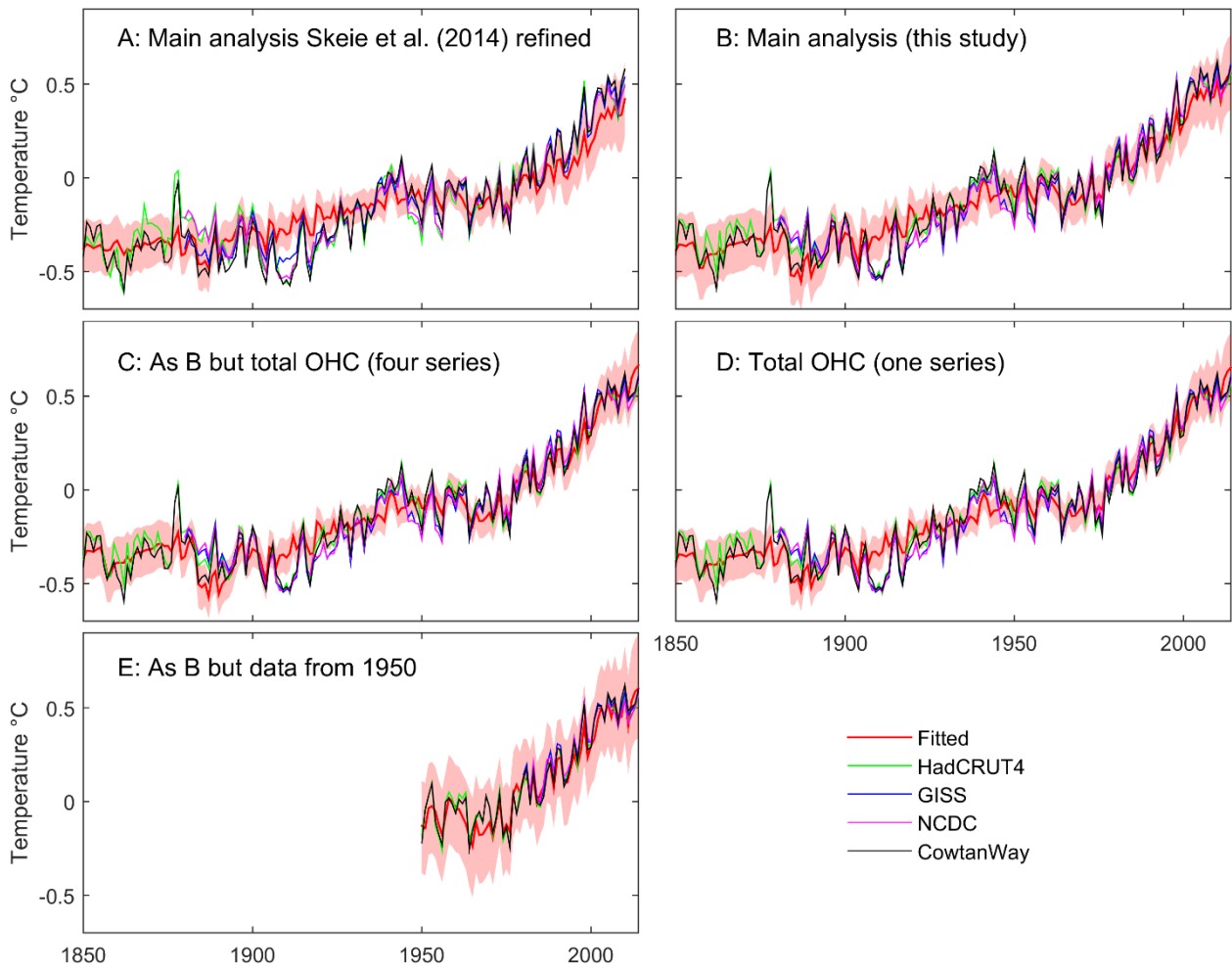

**Figure 2: Observed and fitted (posterior mean) values for the GMST. The shaded areas show the 90% CI for fitted values i.e. the sum of the output from the deterministic SCM and the short-term internal variability excluding the terms for long-term internal variability and model error. Fig. S3 show three set of fitted values for the GMST for the main analysis that include the long-term internal variability and model error.**

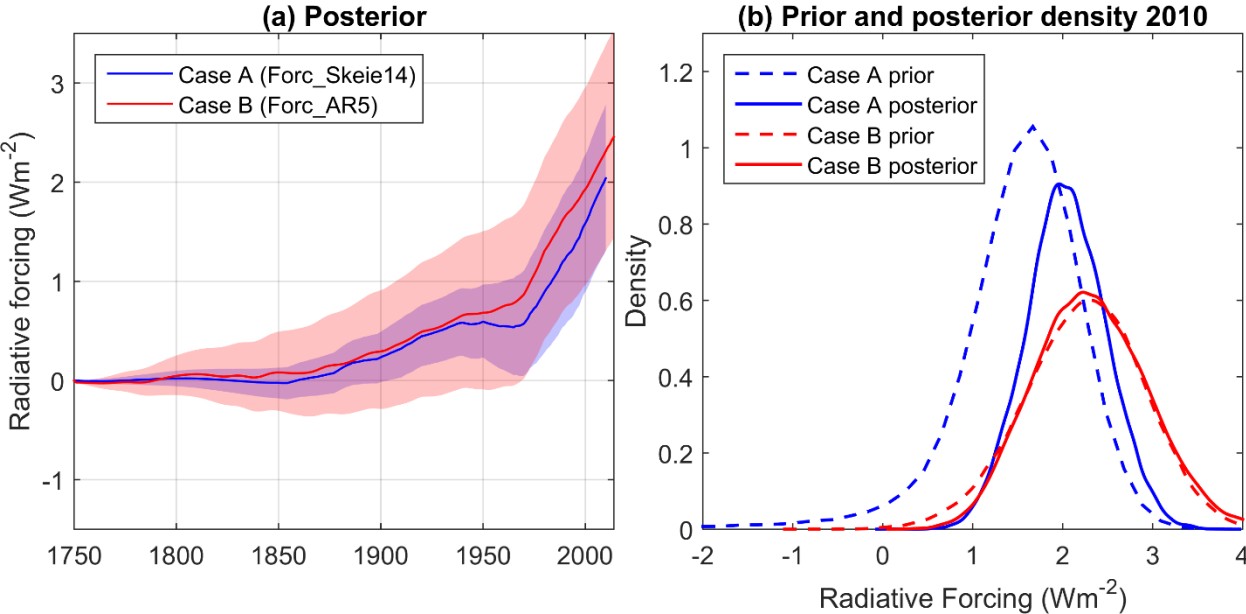

**Figure 3: Posterior distribution of time series (a) and prior (dotted) and posterior (solid) probability density function (PDF) in 2010 (b) for anthropogenic forcing. The shaded areas in (a) represent the 90% CI.**

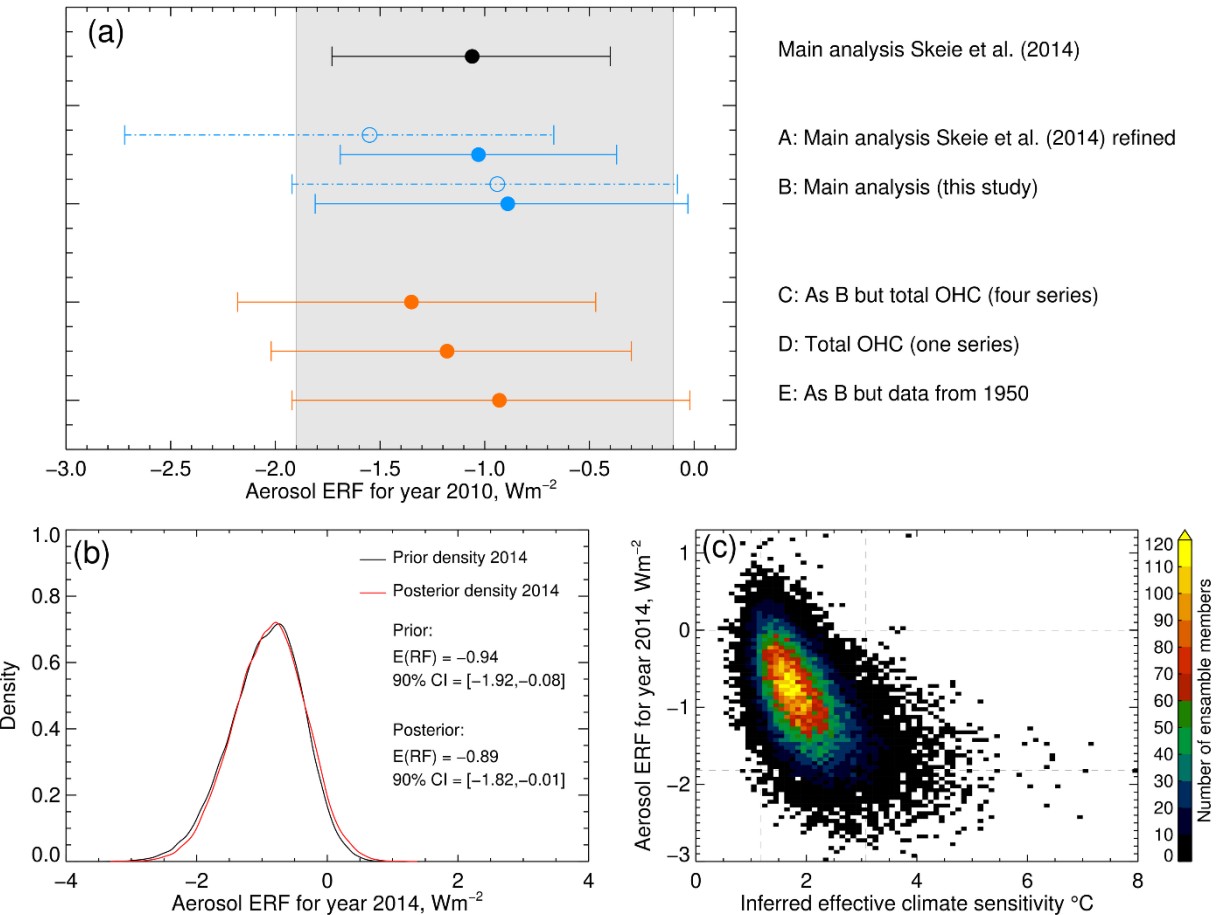

**Figure 4: Posterior 90% CI for aerosol ERF in 2010 for the different analyses in this study (a). The estimated posterior mean is indicated by a dot. The two set of priors used is shown as dash-dotted bars with mean value as an open circle. The IPCC AR5 90% probability range for aerosol ERF is presented as gray shadings. The prior and posterior PDF of RF in 2014 the total aerosol effect in case B (b). Red color for the posterior distributions and black lines for the prior distribution. Panel c) show the relationship between $ECS_{inf}$ and aerosol ERF for case B. The posterior 90% CI is indicated by dashed lines.**

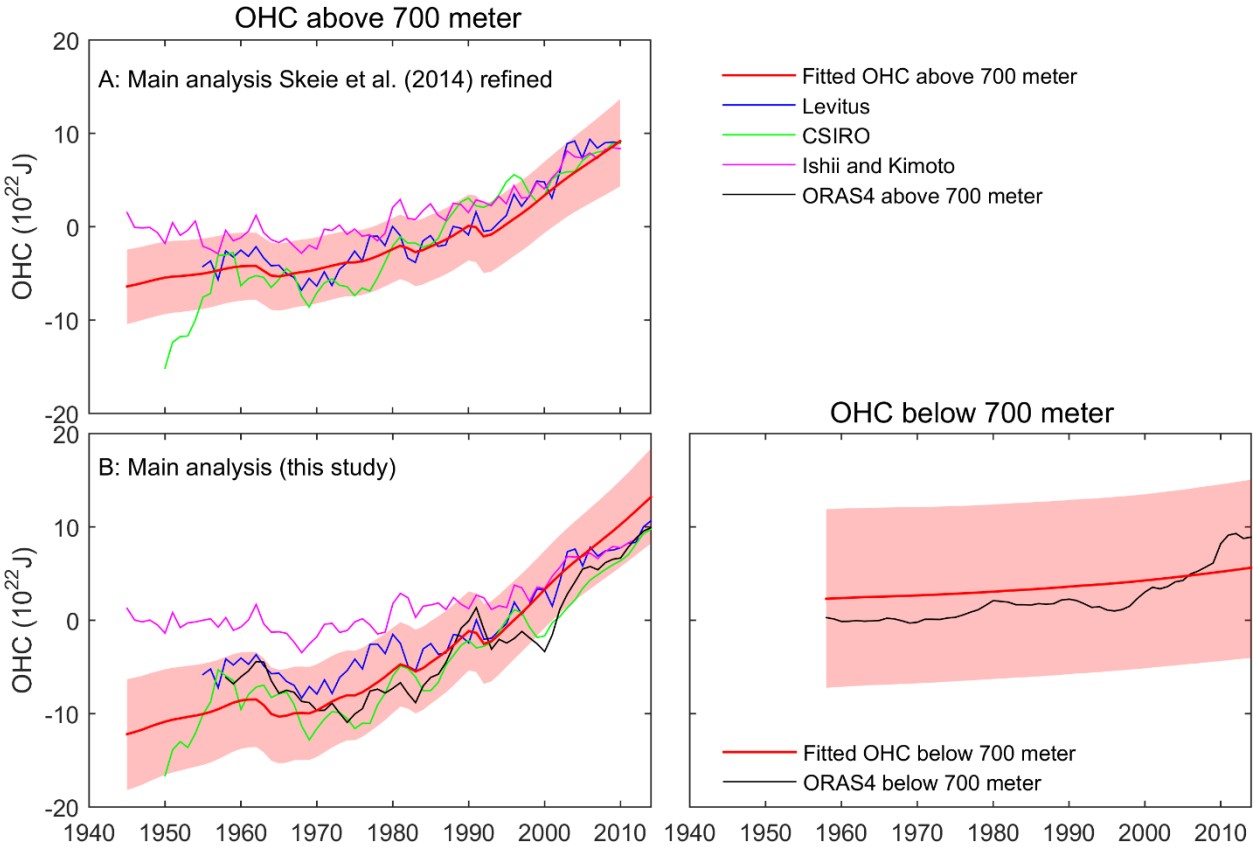

**Figure 5: Observed and fitted (posterior mean) values for the OHC. The shaded areas indicate the 90% CI. Left column: Upper 700m. Right column: Below 700m, if data included in the analysis.**

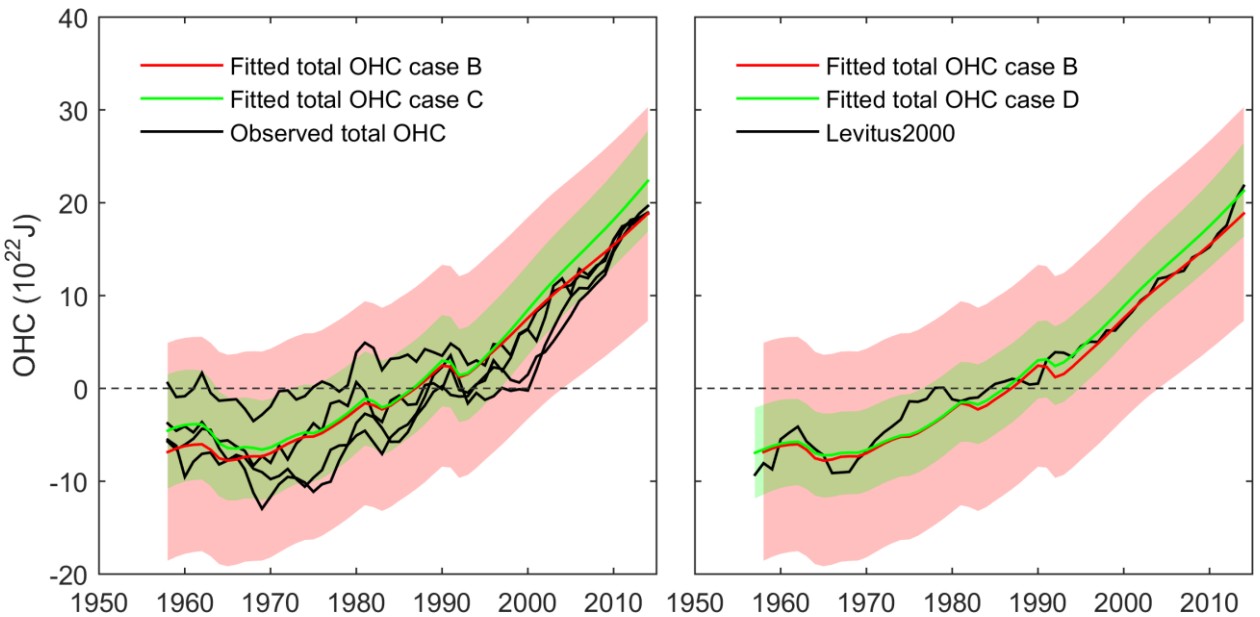

**Figure 6: Observed and fitted (posterior mean) total OHC using several OHC dataset (case B: separate OHC data above and below 700m and C: merge OHC data above and below 700m, left panel) and using only one dataset for the total OHC (case D, right panel). The shaded areas indicate the 90% CI.**

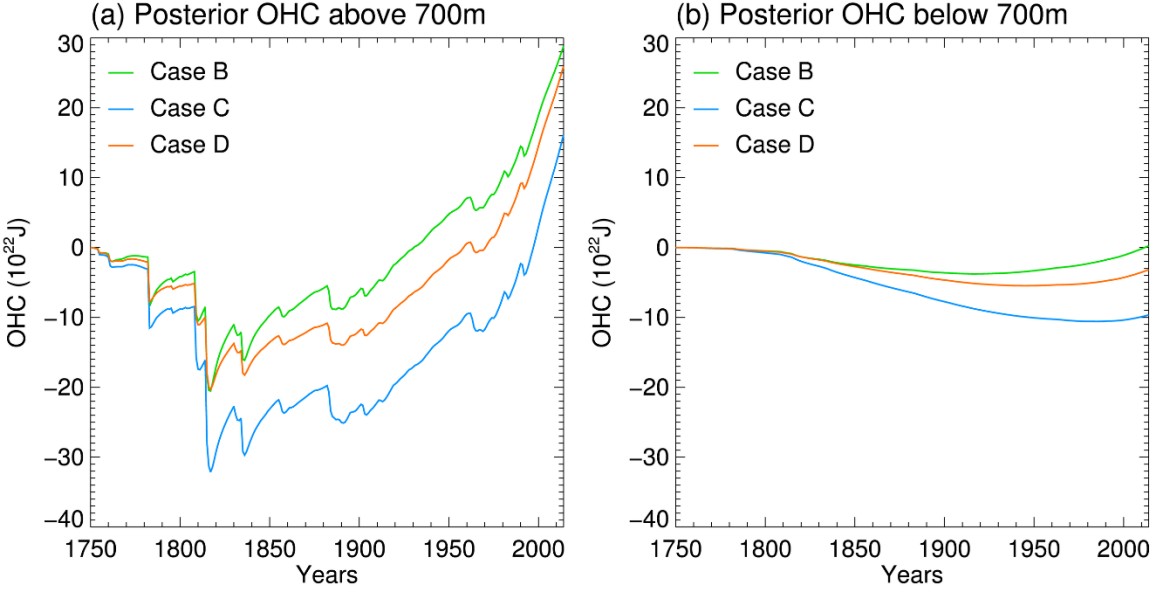

**Figure 7: Posterior mean (solid lines) of the output from the deterministic SCM for OHC above 700m (a) and below 700m (b) for case B, C (total OHC four series) and D (total OHC one series).**

**Table 1: List of data used in the estimation, the abbreviation used in the text, references, in which cases the datasets are used and time of download. The months in parentheses are when data used in case A (see sect. 2.2) were downloaded.**

| Abbreviation | References | Dataset used in case | Downloaded |
|---|---|---|---|
| **Surface temperature change:** | | | |
| GISS | (Hansen et al., 2006;Hansen et al., 2010) | A, B, C, D, E | March 2015 (March 2011) |
| HadCRUT4 | (Morice et al., 2012) | A, B, C, D, E | March 2015 (March 2011[*]) |
| NCDC | (Smith and Reynolds, 2005;Smith et al., 2008) | A, B, C, D, E | March 2015 (June 2011) |
| CowtanWay | (Cowtan and Way, 2014) | A, B, C, D, E | March 2015 (April 2014) |
| **Ocean heat content upper 700 meters:** | | | |
| Levitus | (Levitus et al., 2009) | A, B, C, E | March 2015 (March 2011) |
| CSIRO | (Domingues et al., 2008;Church et al., 2011) | A, B, C, E | April 2014 (October 2011) |
| Ishii and Kimoto | (Ishii and Kimoto, 2009) | A, B, C, E | March 2015 (October 2011) |
| ORAS4 | (Balmaseda et al., 2013) | B, C, E | March 2015 |
| **Ocean heat content below 700 meters:** | | | |
| ORAS4 | (Balmaseda et al., 2013) | B, C, E | March 2015 |
| **Ocean heat content above 2000 meters:** | | | |
| Levitus2000 | (Levitus et al., 2012) | D | July 2015 |
| **SOI-index:** | | | |
| SOI | Southern Oscillation index, Bureau of Meteorology, Australia http://www.bom.gov.au/climate/current/soihtm1.shtml | A, B, C, D, E | March 2015 (November 2011) |
| **Forcing time series:** | | | |
| Forc_Skeie14 | (Skeie et al., 2011;Skeie et al., 2014) | A | |
| Forc_AR5 | (Myhre et al., 2013) | B, C, D, E | |

[*] HadCRUT3

**Table A1: Variance ratios $Var_{obs}/\text{Var}_{rep}$ for pairwise differences of OHC series.**

| OHC series 1 | OHC series 2 | $Var_{obs}/\text{Var}_{rep}$ |
|---|---|---|
| CSIRO | Levitus | 0.21 |
| CSIRO | Ishii and Kimoto | 0.43 |
| CSIRO | ORAS4 | 0.17 |
| Levitus | Ishii and Kimoto | 2.13 |
| Levitus | ORAS4 | 3.74 |
| Ishii and Kimoto | ORAS4 | 3.49 |

