# Peer review of "Climate sensitivity estimates – sensitivity to radiative forcing time series and observational data"

_Earth System Dynamics, 2017_

## Referee Comment (RC1) · Anonymous Referee #1 · 23 Jan 2018

The paper "Climate sensitivity estimates - sensitivity to radiative forcing time series and observational data" (by R.B. Skeie, T. Berntsen, M. Aldrin, M. Holden, and G. Myhre) is a sensitivity study on the use of different observational datasets to infer estimates of the effective climate sensitivity using an energy balance model in a Bayesian framework. The paper is essentially a refinement of the results of a previous publication, including a systematic analysis of how the use of different data sources impacts the estimates obtained with the model. The paper is of interest from a methodological point of view. The presentation of the methods and of the results is however very confusing. Overall the paper is suited for publication in Earth System Dynamics, after a substantial revision of the presentation of the methodology and of the results.

[Figure]

The authors employ an energy balance climate/upwelling diffusion ocean model, giving as output the surface temperature in the two hemispheres and the global ocean heat content. The model is combined with a stochastic model representing long and short term variability as well as model errors. The equilibrium climate sensitivity is a parameter of the deterministic model, and is constrained by observations in a Bayesian inference. The authors use several observational datasets, including radiative forcings, global surface temperatures, and ocean heat content, performing a fairly systematic analysis of the role of each data source in determining the final results.

The authors have used the same method in a previous publication (Skeie & al 2014) with partially different data. The estimate of the climate sensitivity and its confidence interval change by 5-10%, remaining well inside the range of values obtained with other methods, including the range given by the IPCC report obtained running complex general circulation models. Such a minor change in the main estimate might question the relevance of the new results. However, the authors show in detail how each of the different datasets included or modified with respect to Skeie & al (2014) contribute to determine the final estimate, and how compensations between changes of different sign lead to an overall small final effect. Particular attention is given to the role of the ocean heat content. The paper is therefore of interest for readers working with this kind of methods. The method has been used previously in other publications and I have no major scientific criticisms, a part from a few questions which I include in a bullet list below.

My main criticism to this paper regards the presentation of the methods and of the results. In order to find informations which are essential to have even a minimal understanding of what the authors describe in the main text, the reader is systematically asked to go back and forth between appendixes, supplementary materials, and the authors' previous publication history. As a result, the paper in its current form is extremely hard to follow. For example, no real description is given of the energy balance model. The reader is referred to Skeie & al (2014) and/or Aldrin & al (2012), and even

[Figure]

there the informations are fragmented and partially referring to older publications. And I am not talking about the details of the model. For example, in the description of the model/methods I could not find a qualitative description of how the components of the parameter vector $\theta$ are included in the model. However, some of these parameters are later discussed in the paper, out of the blue for a reader who has not worked with this specific model. This is just an example, there are many others. The same holds for the way the deterministic model is combined with the stochastic terms, and many other aspects of the procedures followed by the authors to obtain their results.

While it is clear that the technical details of the models and of the methodology can (must) be left out of the main text of the paper, in particular if they have been described elsewhere, a minimal but clear and comprehensive description of the models and methods must be present in the paper. To the maximum extent possible, the paper has to be readable stand alone. A similar point holds for the use of appendixes and supplementary materials. They should be used to provide technical details not necessary to follow the flow of the main text, or figures giving complementary informations. Instead, in the way the authors use them, there is no logical difference between figures included in the main text and figures included in the supplementary informations, and in order to understand what the authors have done (again, not the details: the very procedure) it is often necessary to stop reading, move to an appendix, and then come back to to main text. This is extremely confusing, and makes the paper unnecessarily hard to read.

The authors should revise their paper in order to make it clear and readable. A simple but comprehensive description of the models and methods they use that are not standard techniques should be provided. The interplay between the main text and appendixes and supplementary materials should be simplified. That said, I have some more specific remarks which I list below.

1. Page 2, lines 3-6. "Since the current climate is in a non-equilibrium state observationally based methods can only account for the feedbacks operating during the historical period. Thus, these estimates are often referred to as inferred or effective climate sensitivity (Armour, 2017;Forster, 2016) and are generally significantly lower than ECS estimates from Atmosphere-Ocean General Circulation Models (AOGCMs).". Just a comment on this. This is an important remark and I agree with the authors in stressing the difference between "real" equilibrium climate sensitivity and inferred climate sensitivity. Actually, this can be seen rigorously and generally in response theory of dynamical systems. In this framework the ECS can be written as a weighted integral of the imaginary part of linear susceptibility of the system over all frequencies, which implies that one needs all time scales in order to correctly compute the ECS. A similar result holds for transient definitions of the climate sensitivity, like the Transient Climate Response. The authors can find a discussion on this for example in Ragone & al (2016) (equations 9 and 14) and Lucarini & al (2017);

2. Page 3, lines 1-7. Here it would be good to give a (very) brief description of energy balance based estimates, of the peculiarities of the method developed by the authors, how they have used it in the past and which results they have obtained, and what the current paper adds to these previous works. This is somehow already done, but it should be more clear and systematic;

3. Page 3, lines 10-28. The description of the model and methods should be expanded and made clear;

4. Page 3, line 21-23. "Most of the data series are provided with corresponding yearly standard errors. However, these are often small compared to the differences between the data series, indicating that the errors reported by the data providers are too small". This is an interesting observation by the authors, and I agree with them that in this situation using only one dataset would result in

assuming uncertainties that are probably too small. However, to say that data providers provide errors that are too small is quite a statement. If the authors are aware of a discussion in the literature on the statistical non consistency between different datasets of the quantities they refer to, it would be good to be more specific on which datasets are in disagreement with each other and to include some references;

5. Section 2.1. The results about the Transient Climate Sensitivity are actually interesting. If the authors find it possible, I would include them in the main text and discuss them a bit more;

6. Section 3. Forc_Skeie2014 and Forc_AR5 have never been defined;

7. Page 4, lines 29-32. Why the match between prior and posterior distributions in figure 3 changes so much between cases A and B? The authors somehow discuss this in lines 6-12 of page 5, but the change is impressive. Can the authors say something more about this? Note also that a clear definition of the priors is somewhat missing in this paper (the one of the equilibrium climate sensitivity for example is not mentioned at all). The authors probably assume that the reader should go looking for it in Skeie & al (2014), which I did, but it is one of those things that should be repeated also in this paper when presenting the methods;

8. Page 5, lines 25-30. The data from Ishii and Kimoto are completely out of the confidence interval, in particular in case B. Can the authors comment about this?

9. Page 6, lines 22-23. "... and hence no reason to refine the IPCC 2013 aerosol ERF best estimate jet.", there is something wrong with this sentence;

10. Page 7, lines 8-14. If the authors had used the same errors as in Johansson & al (2015), how would the range of climate sensitivity differ? In other words, how much of the larger range is due to considering larger errors and how much to the differences between their methods? Can they comment on this here?

[Figure]

Beside this, there are a number of typos and errors that should be taken care of. For example, use CI instead of C.I., consistently with the use of other acronyms. Please revise the paper also from this point of view.

References

Ragone F., Lucarini V., Lunkeit F., A new framework for climate sensitivity and prediction: a modelling perspective, Clim. Dyn., 46(5-6), 1459-1471 (2016);

Lucarini V., Ragone F., Lunkeit F., Predicting climate change using response theory: global averages and spatial patterns, J. Stat. Phys., 166(3-4), 1036-1064 (2017).

---

## Short Comment (SC1) · 29 Jan 2018

In my view the article, while in principle suitable for publication by ESD, would be much improved if the following issues were addressed:

1. The information provided on the results is too limited. The median is a more appropriate best estimate measure than the mean for skewed distributions such as that for ECS. That is why the IPCC AR5 report gave medians, but not means, for all the observationally-based ECS estimates that it showed (Figure 10.20b). The medians should be shown, at least for the ECS and TCR posteriors, either instead of or in addition to the means, and likewise given in the Abstract and the main text. It is also slightly

strange, for a Bayesian analysis, that the posterior PDF for ECS is only shown in panel j) of Figure S1.

2. The TCR estimate is of interest to readers as well as the ECS estimate, but it only seems to feature in Figure S2, with no values given. The Main analysis median TCR estimate and its 5-95% uncertainty range should be stated in the main text and, preferably, also in the Abstract.

3. The study uses a subjective Bayesian analysis. The priors used likely have a major influence on the results, but finding out what they are requires referring to both Skeie et al 2014 and Aldrin et al 2012. A wide uniform prior seems to be used for ECS. It is well known that doing so biases ECS estimation upwards and greatly fattens the upper tail of the posterior. (Annan and Hargreaves 2011: "We show that the popular choice of a uniform prior has unacceptable properties and cannot be reasonably considered to generate meaningful and usable results."; Lewis 2014). Using a noninformative joint prior would produce estimates that were at least approximately unbiased, but calculating one could be difficult. Providing results based purely on the joint likelihood function, using a frequentist profile likelihood method, would be a reasonable alternative. If, as seems likely to be the case, the profile likelihood peaks at approximately the same point as the marginal likelihood for ECS (being the mode of the posterior, as a uniform prior for ECS is used) then the maximum likelihood estimate for ECS would be ∼1.75 K.

Also, showing what the characteristics of the ECS posterior are when a prior for ECS that is uniform in 1/ECS (and therefore is proportional to 1/ECS^2) is used would be helpful. That prior will be close to noninformative. [Given that fractional uncertainty in forcing (RF) is approximately symmetrical (Fig. 3(b)) and dominates that in GMST (and in ocean heat uptake), a uniform prior will be approximately noninformative for 1/ECS, and on a change of variable to ECS a uniform prior becomes 1/ECS^2.]

4. The stepwise update results are interesting, but difficult to interpret in the absence

of adequate quantitative information as to the changes in data values and uncertainty ranges involved.

Nicholas Lewis

References

J. D. Annan and J. C. Hargreaves (2011) On the generation and interpretation of probabilistic estimates of climate sensitivity Climatic Change 104, 423–436

N Lewis (2014) Objective Inference for Climate Parameters: Bayesian, Transformation-of-Variables, and Profile Likelihood Approaches. J Climate 27, 7270-7284

---

## Referee Comment (RC2) · Anonymous Referee #2 · 26 Feb 2018

This paper is an updated analysis of Skeie et al., 2014, using the same simple model and analysis method, but updated radiative forcing (following IPCC AR5), deep ocean data, and updated data to 2014. The paper is very well written and carefully executed. Its a very valuable addition to the literature, and sheds light to the effect of various analysis choices and data uncertainties on the overall ECS result. I also like the discussion of the Johannsen result in comparison to the paper here, which does favour substantially higher ECS values. it is very informative to see this discussion and I find this aspect of the paper particularly valuable.

There are some aspects of the paper that I think could still be improved: a) I find the

treatment of gaps in observations opaque. I THINK what happens is that the observations are treated as an estimate of global mean temperature i.e. the fully covered simple model is compared to the observations which do contain missing data. This needs to be clarified. It also affects the results of Richardson et al which are as strong as cited only for the case of comparing a fully covered model with gappy data. If accounting for the actual observational coverage by comparing like with like (model limited to same datapoints), the result gets less sensitive which is reasonable. Its not necessary to change the method, but it is necessary to be clear please. If the full model field is compared to the data, it might also be useful to be more explicit about the lack of coverage in rapidily warming high latitudes. For the deep ocean, it seems that the upper 700m are compared in model and data for the upper ocean case only - this could be again said more explicitly

b) prior: while I am not as convinced that the nonanonymous reviewer about the value of an objective prior, I did like that earlier papers of this group showed results using different priors. I think it would be nice to do this here as well in order to illustrate to what extent the prior still matters.

c) the plotted updated Skeie 2014 analysis is different from the published version - I would find it cleaner to also add the originally published range in figure 1.

d) it would be nice to see a bit more about the TCR results here as well.

e) The sensitivity tests are interesting and it would be useful to discuss some of them in the body of the text, particularly the case of efficacy - maybe even give some further ranges in abstract. There is clearly some sensitivity here that isnt accounted for in the main result, so this should be made more clear.

f) it is not quite clear to me how internal climate variability is treated. I assume it is done as in the main Skeie et al., 2014 result, i.e. assuming that internal variability is reflected in the residual and no additional estimate is given. It would be informative to hear how this estimate compares for example to model estimates. It is interesting to

hear that the updated RF series yields smaller residuals.

g) should we treat the main analysis B as the more reliable result or the total OHC using 4 series case C? its is quite remarkable how much the pdf shifts in the latter case. Is the model good enough to reliably separate upper and lower ocean? there is some good discussion in the paper but I am left undecided about this. A bit more clarity would be helpful.

Small comments: p 3 model section: should this refer here already to using the Andronova and Schlesinger model? also, what are the 7 parameters? this would be very useful to hear what they parameterize and which parameter uncertainties are systematically investigated and which are not. this is probably retrievable from earlier papers but worth reiterating here.

Figure 5 discussion and caption: It would be helpful for a reader to understand what case B,C,D are from the caption - I got it on second but not first reading. (eg averaging rather than separate treatment etc).

In figure 4, I find it slightly confusing that one of the OHC series is systematically outside the estimated 90% range. can this be explained in the text?

---

## Author Comment (AC1) · 23 Mar 2018

We appreciate the useful comments by both reviewers and the comments made by N. Lewis on the manuscript "Climate sensitivity estimates – sensitivity to radiative forcing time series and observational data". Below follow our responses to the comments by the reviewers and the short comment, as well as descriptions of how the manuscript has been modified. The original reviewer's comments are in black and our response in blue.

**Anonymous Referee #1**

The paper "Climate sensitivity estimates - sensitivity to radiative forcing time series and observational data" (by R.B. Skeie, T. Berntsen, M. Aldrin, M. Holden, and G. Myhre) is a sensitivity study on the use of different observational datasets to infer estimates of the effective climate sensitivity using an energy balance model in a Bayesian framework. The paper is essentially a refinement of the results of a previous publication, including a systematic analysis of how the use of different data sources impacts the estimates obtained with the model. The paper is of interest from a methodological point of view. The presentation of the methods and of the results is however very confusing. Overall the paper is suited for publication in Earth System Dynamics, after a substantial revision of the presentation of the methodology and of the results.

The authors employ an energy balance climate/upwelling diffusion ocean model, giving as output the surface temperature in the two hemispheres and the global ocean heat content. The model is combined with a stochastic model representing long and short term variability as well as model errors. The equilibrium climate sensitivity is a parameter of the deterministic model, and is constrained by observations in a Bayesian inference. The authors use several observational datasets, including radiative forcings, global surface temperatures, and ocean heat content, performing a fairly systematic analysis of the role of each data source in determining the final results.

The authors have used the same method in a previous publication (Skeie & al 2014) with partially different data. The estimate of the climate sensitivity and its confidence interval change by 5-10%, remaining well inside the range of values obtained with other methods, including the range given by the IPCC report obtained running complex general circulation models. Such a minor change in the main estimate might question the relevance of the new results. However, the authors show in detail how each of the different datasets included or modified with respect to Skeie & al (2014) contribute to determine the final estimate, and how compensations between changes of different sign lead to an overall small final effect. Particular attention is given to the role of the ocean heat content. The paper is therefore of interest for readers working with this kind of methods. The method has been used previously in other publications and I have no major scientific criticisms, a part from a few questions which I include in a bullet list below.

My main criticism to this paper regards the presentation of the methods and of the results. In order to find informations which are essential to have even a minimal understanding of what the authors describe in the main text, the reader is systematically asked to go back and forth between appendixes, supplementary materials, and the authors' previous publication history. As a result, the paper in its current form is extremely hard to follow. For example, no real description is given of the energy balance model. The reader is referred to Skeie & al (2014) and/or Aldrin & al (2012), and even there the informations are fragmented and partially referring to older publications. And I am not talking about the details of the model. For example, in the description of the model/methods I could not find a qualitative description of how the components of the parameter vector _ are included in the model. However, some of these parameters are later discussed in the paper, out of the blue for a reader who has not worked with this specific model. This is just an example, there are many others. The same holds for

the way the deterministic model is combined with the stochastic terms, and many other aspects of the procedures followed by the authors to obtain their results.

While it is clear that the technical details of the models and of the methodology can (must) be left out of the main text of the paper, in particular if they have been described elsewhere, a minimal but clear and comprehensive description of the models and methods must be present in the paper. To the maximum extent possible, the paper has to be readable stand alone. A similar point holds for the use of appendixes and supplementary materials. They should be used to provide technical details not necessary to follow the flow of the main text, or figures giving complementary informations. Instead, in the way the authors use them, there is no logical difference between figures included in the main text and figures included in the supplementary informations, and in order to understand what the authors have done (again, not the details: the very procedure) it is often necessary to stop reading, move to an appendix, and then come back to to main text. This is extremely confusing, and makes the paper unnecessarily hard to read.

The authors should revise their paper in order to make it clear and readable. A simple but comprehensive description of the models and methods they use that are not standard techniques should be provided. The interplay between the main text and appendixes and supplementary materials should be simplified. That said, I have some more specific remarks which I list below.

There is always a balance between the length of the paper and what must be included in the main text. We can agree with the reviewer that in this case too much has been omitted and put into the appendix, as supplementary material or just referred to previous work.

The data and method section is significantly expanded to include the essential information needed for the readers to be able to understand our method and follow the discussion. Still some of the information has to be retained in the appendices, but we believe the reorganization should satisfy the request from the reviewers. This is described under points 1-3 below.

To simplify the interplay between the main text and the appendices, "Appendix A: Refinement of Skeie14" and "Appendix B: Extending data up to and including 2014" are kept as appendices while the others are merged with the main text. In addition, Figure S2 (the TCR figure) and Figure S6 as well as bottom right panel of Figure S7b are now included in the main text to make the manuscript easier to follow.

1. Page 2, lines 3-6. "Since the current climate is in a non-equilibrium state observationally based methods can only account for the feedbacks operating during the historical period. Thus, these estimates are often referred to as inferred or effective climate sensitivity (Armour, 2017;Forster, 2016) and are generally significantly lower than ECS estimates from Atmosphere-Ocean General Circulation Models (AOGCMs).". Just a comment on this. This is an important remark and I agree with the authors in stressing the difference between "real" equilibrium climate sensitivity and inferred climate sensitivity. Actually, this can be seen rigorously and generally in response theory of dynamical systems. In this framework the ECS can be written as a weighted integral of the imaginary part of linear susceptibility of the system over all frequencies, which implies that one needs all time scales in order to correctly compute the ECS. A similar result holds for transient definitions of the climate sensitivity, like the Transient Climate Response. The authors can find a discussion on this for example in Ragone & al (2016) (equations 9 and 14) and Lucarini & al (2017);

Thank you for pointing at these studies. We will look into this theoretical framework to see how it may be used for our future work.

2. Page 3, lines 1-7. Here it would be good to give a (very) brief description of energy balance based estimates, of the peculiarities of the method developed by the authors, how they have used it in the past and which results they have obtained, and what the current paper adds to these previous works. This is somehow already done, but it should be more clear and systematic;

We will modify the text to the following:
"In this study we use our estimation model that were first documented in Aldrin et al. (2012) and further developed in Skeie et al. (2014). Our method is more complex than the common energy balance based estimates (Forster, 2016) in that we embed a simple climate model into a stochastic model with radiative forcing time series as input, treating the NH and SH separately and includes a vertical resolution of the ocean (40 layers). The radiative forcing time series are linked to the observations of OHC and temperature change through the simple climate model and the stochastic model, using a Bayesian approach. A unique feature with our method is that we use several observational datasets. The method estimates not only the $ECS_{inf}$ but simultaneously also provides posterior estimates of the radiative forcing, as well as posterior uncertainty estimates in the observations datasets and correlations between them. In this study we further develop our estimation model with additional observational datasets, including heating rates of the deep ocean (below 700m) and new forcing time series from the IPCC AR5 as well as extended time series from 2010 to 2014 to update our estimate of $ECS_{inf}$. We carry out a number of sensitivity experiments to investigate causes of differences in observational based $ECS_{inf}$ estimates due to differences in the input data (observations of surface temperature, OHC and RF)."

3. Page 3, lines 10-28. The description of the model and methods should be expanded and made clear;

In the model section we have included a paragraph on the SCM before we present $g_t = m_t(x_{1750:t}, \theta) + n_t$ . The $n_t$ term is then described in more detail (see the response to reviewer 2). At the end of the method section the priors are presented (see response below).

The following section describing the SCM is included:

"The core of our model framework is the SCM, a deterministic energy balance/upwelling-diffusion model (Schlesinger et al., 1992). The SCM calculates annual hemispheric and global mean near-surface temperature change (blended SST and surface air temperature) and changes in global OHC as a function of estimated RF time series. The vertical resolution of the ocean is 40 layers. The output of the SCM can be written as $m_t(x_{1750:t}, \theta)$, where $x_{1750:t}$ (the RF from 1750 until year t) and $\theta$ are the true, but unknown, input values to the SCM. $\theta$ is a vector of seven parameters, each with a physical meaning. One of these parameters is the climate sensitivity, and other parameters determine how the heat is mixed into the ocean, which includes the mixed layer depth, the air-sea heat exchange coefficient, the vertical diffusivity in the ocean and the upwelling velocity (see Schlesinger et al. (1992) and Aldrin et al. (2012) for details)."

4. Page 3, line 21-23. "Most of the data series are provided with corresponding yearly standard errors. However, these are often small compared to the differences between the data series, indicating that the errors reported by the data providers are too small". This is an interesting observation by the authors, and I agree with them that in this situation using only one dataset would result in assuming uncertainties that are probably too small. However, to say that data providers provide errors that are too small is quite a statement. If the authors are aware of a discussion in the literature on the statistical non consistency between different datasets of the quantities they refer to, it would be good to be more specific on which datasets are in disagreement with each other and to include some

references;

We have not seen that this has been discussed in the literature, except that in one of our previous papers. We will present the results from a small analysis where we compare the reported uncertainties with the differences between the corresponding data series, both for temperatures and OHC. This will be included as a table in the supplementary material.

5. Section 2.1. The results about the Transient Climate Sensitivity are actually interesting. If the authors find it possible, I would include them in the main text and discuss them a bit more;

The TCR results are included as an additional panel in Fig. 1 as well as we provide numbers in the main text.

6. Section 3. Forc_Skeie2014 and Forc_AR5 have never been defined;

They are defined in table 1, but the definition will now also be included in the main text. In section 2.1 we now include: "The forcing time series are hereafter named Forc_Skeie2014 and the priors of each forcing mechanisms included (Table S1) are described in detail in the appendix D of Skeie14."

At the beginning of section 3 we replace the sentence: "We replaced the Forc_Skeie14 with the AR5 effective radiative forcing (ERF) estimates (Myhre et al., 2013), including the AR5 uncertainties" with "We replaced the Forc_Skeie14 with the AR5 effective radiative forcing (ERF) estimates (Myhre et al., 2013) hereafter named Forc_AR5. The priors for the forcing mechanisms included (Table S1) are constructed to be consistent with the uncertainties provided in AR5 and the same relative uncertainty for the prior forcing is used over the entire time period.

We hope these two lines also satisfy part of the reviewers next comment regarding the definition of the (RF) priors.

7. Page 4, lines 29-32. Why the match between prior and posterior distributions in figure 3 changes so much between cases A and B? The authors somehow discuss this in lines 6-12 of page 5, but the change is impressive. Can the authors say something more about this?

We have added a short comment on this in the text at line 30, page 4 in the previous version:

"The prior anthropogenic mean forcing in 2010 increased from 1.5 to 2.3 Wm-2 from case A to case B when Forc_AR5 replaced Forc_Skeie14. For case A, the posterior forcing is shifted upwards compared to the prior, suggesting that the data supports higher values than the Forc_Skeie14 prior used in this case. When the prior is changed to Forc_AR5 in case B, the posterior is much closer to the prior which indicates that the data is more in accordance with the new prior than the old one."

The most uncertain forcing mechanism is the aerosol ERF, and we also discuss the aerosol ERF prior and posterior in more detail and therefore have we added a new figure number 4, with a panel similar to figure 1 but for aerosol ERF in year 2010, a panel with prior and posterior pdf for the aerosol ERF for case B and a panel for a scatter density plot for aerosol ERF and $ECS_{inf}$ for case B.

Note also that a clear definition of the priors is somewhat missing in this paper (the one of the equilibrium climate sensitivity for example is not mentioned at all). The authors probably assume that the reader should go looking for it in Skeie & al (2014), which I did, but it is one of those things that should be repeated also in this paper when presenting the methods;

We have added the priors for the ECS in the methods section and included a table in the supplementary material for the informative priors for the other parameters in the SCM.

In the method section we now include: "The unknown quantities are given prior distributions as presented in Skeie et al. (2014). The ECS is given a vague prior, uniform (0,20) and the informative priors for $\theta$ based on expert judgment are listed in Table S2. "

8. Page 5, lines 25-30. The data from Ishii and Kimoto are completely out of the confidence interval, in particular in case B. Can the authors comment about this?

Reviewer 2 has the same comment. We have added the following sentence to the manuscript:

"Remark that the Ishii and Kimito series is out of the 90% CI. The reason is that the assumed observational errors for all series are much larger back in time than in the recent years (Appendix A). Therefore the various data series are aligned quite close to each other in the recent years, and since the Ishii and Kimoto series has a much weaker trend than the others, it lies above the 90% CI in the first part of the data history."

9. Page 6, lines 22-23. "... and hence no reason to refine the IPCC 2013 aerosol ERF best estimate jet.", there is something wrong with this sentence;

We have replaced the sentence by "These recent studies indicate that there is no reason to refine the IPCC 2013 aerosol ERF best estimate yet."

10. Page 7, lines 8-14. If the authors had used the same errors as in Johansson & al (2015), how would the range of climate sensitivity differ? In other words, how much of the larger range is due to considering larger errors and how much to the differences between their methods? Can they comment on this here?

This is a nice suggestion for a further study, preferentially in a multimodel study. As we stated in the text it would have been of great value to do a multi-model intercomparison of observational methods using identical input data to investigate the uncertainties due to different models.

Beside this, there are a number of typos and errors that should be taken care of. For example, use CI instead of C.I., consistently with the use of other acronyms. Please revise the paper also from this point of view.

All C.I. are replaced by CI and some other small typos are corrected.

**Anonymous Referee #2**

This paper is an updated analysis of Skeie et al., 2014, using the same simple model and analysis method, but updated radiative forcing (following IPCC AR5), deep ocean data, and updated data to 2014. The paper is very well written and carefully executed. Its a very valuable addition to the literature, and sheds light to the effect of various analysis choices and data uncertainties on the overall ECS result. I also like the discussion of the Johannsen result in comparison to the paper here, which does favour substantially higher ECS values. it is very informative to see this discussion and I find this aspect of the paper particularly valuable.

There are some aspects of the paper that I think could still be improved: a) I find the treatment of gaps in observations opaque. I THINK what happens is that the observations are treated as an estimate of global mean temperature i.e. the fully covered simple model is compared to the observations which do contain missing data. This needs to be clarified. It also affects the results of Richardson et al which are as strong as cited only for the case of comparing a fully covered model with gappy data. If accounting for the actual observational coverage by comparing like with like (model

limited to same datapoints), the result gets less sensitive which is reasonable. Its not necessary to change the method, but it is necessary to be clear please. If the full model field is compared to the data, it might also be useful to be more explicit about the lack of coverage in rapidly warming high latitudes. For the deep ocean, it seems that the upper 700m are compared in model and data for the upper ocean case only - this could be again said more explicitly

The SCM calculates hemispheric temperatures for blended SST and SAT. For observed surface temperature we use several data series, some with a greater global coverage than others. For instance HadCRUT4 with more gaps, compared to CowtanWay with a larger degree of infilling. However, considering the whole time period, there are also large uncertainties related to these infilling methods. Since our SCM does not have a spatial grid structure, we cannot filter out regions that have very sparse observations. Thus in the model we have chosen to use them all, with the interpretation that they represent the hemispheric temperatures of the SCM. In the introduction we insert a sentence regarding the gappy data to highlight this point: "Several observed surface temperature records exist with different methods to account for spatial gaps in the observations".

In the method section we have include a short description of the SCM, and specify that it calculate hemispheric temperatures for blended SST and SAT, se response to reviewer 1 above.

Replying to your last comment, the text describing the sensitivity test regarding the deep ocean data in the appendix is merged with the existing text in section 4.1. We hope this will make it clearer.

b) prior: while I am not as convinced that the nonanonymous reviewer about the value of an objective prior, I did like that earlier papers of this group showed results using different priors. I think it would be nice to do this here as well in order to illustrate to what extent the prior still matters.

N. Lewis has a related comment, see below. To demonstrate the sensitivity of the prior for ECS, we have computed an alternative estimate of ECS using another prior, namely where 1/ECS is uniformly distributed. This is done by importance sampling of the MCMC samples we already have, as described in Appendix E6 in our previous paper Skeie et al. (2014).

"Implementing an alternative prior for $ECS_{inf}$ as in Skeie14, where $1/ECS_{inf}$ is uniformly distributed, the 90% CI shift to lower values (0.97 to 2.5°C). The $ECS_{inf}$ estimate is sensitive to the prior, however this alternative prior is strongly informative towards low climate sensitivities that may not be realistic (Aldrin et al. 2012, Skeie et al. 2014)."

c) the plotted updated Skeie 2014 analysis is different from the published version - I would find it cleaner to also add the originally published range in figure 1.

We have added the original Skeie 2014 range to figure 1.

d) it would be nice to see a bit more about the TCR results here as well.

We will add the TCR results as a separate panel in figure 1 as well as a few more additional sentences in the main text presenting the TCR results.

e) The sensitivity tests are interesting and it would be useful to discuss some of them in the body of the text, particularly the case of efficacy - maybe even give some further ranges in abstract. There is clearly some sensitivity here that isnt accounted for in the main result, so this should be made more clear.

We have moved the text regarding the efficacy to the discussion section in the main body of the text.

"Recently, studies have suggested that assuming equal efficacy for all forcings bias the ECS estimate low (Marvel et al., 2015;Shindell et al., 2015) even when ERFs are used. In our approach, the efficacy is implicitly included in the forcing uncertainty and thus accounted for. However, if we apply a efficacy of 1.5 for ozone, surface albedo, BC on snow and aerosols, which is the efficacy found in the analysis of Shindell (2014), the PDF of the ECS is shifted to larger values, with a 90% CI ranging from 1.2 to 3.7°C."

In the abstract we have now specified that the values and ranges given are from our main analysis, indicating that sensitivity test may give different results.

f) it is not quite clear to me how internal climate variability is treated. I assume it is done as in the main Skeie et al., 2014 result, i.e. assuming that internal variability is reflected in the residual and no additional estimate is given. It would be informative to hear how this estimate compares for example to model estimates. It is interesting to hear that the updated RF series yields smaller residuals.

In the model section we have now included a paragraph explaining the three terms representing variability and model error to make this clearer.

"…where $n_t$ is a stochastic process, with three terms, representing long-term and short-term internal variability and model error. For the short-term internal variability, we use the Southern Oscillation index (Table 1) to account for the effect of ENSO. The term for the long-term internal variability were implemented in Skeie14 and the dependence structure of this term (i.e. correlations over time and between the three elements) is based on control simulations with a GCM from CMIP5 (see Skeie14 for details) but the magnitude is estimated from the data. This term will also represent other slowly varying model errors due to potential limitations of the SCM and forcing time series. The third error term is included to account for more rapidly varying model errors."

We put the estimated amplitudes of the internal variability terms ($n_t$) in context with the results from unforced control simulations with the ESM models participating in CMIP5 (Palmer and McNeall, ERL, 2014). We will add the following in the result section:

"The estimated amplitude of the mulitidecadal internal variability (about 0.2°C in each hemisphere, cf. Figure S5) is in good agreement with the decadal trends in global surface temperatures from unforced control simulations in the multimodel ensemble from CMIP5 (0.2-0.4°C, Palmer and McNeall, 2014)"

g) should we treat the main analysis B as the more reliable result or the total OHC using 4 series case C? its is quite remarkable how much the pdf shifts in the latter case. Is the model good enough to reliably separate upper and lower ocean? there is some good discussion in the paper but I am left undecided about this. A bit more clarity would be helpful.

When OHC above and below 700m is merged, most of the heat is stored in the upper 700m (as shown in Fig S7 which we will move to the main text) while the observations show an increase in OHC in the deeper layers. The aerosol forcing is then allowed to be stronger, and ECS$_{inf}$ is shifted to larger values. The model is simple, and we acknowledge that when this point is discussed in section 5 (page 9 line 1-7 in the original manuscript), however, we believe separating above and below 700m is more realistic and thus case B is our best estimate. Regardless of which estimate is the better, a key finding is that information about the vertical structure of the OHC trends constrains the aerosol forcing and thus provides a potential avenue for improved ECS estimates.

At the end of presenting the results for case C in section 4.2 we add the following sentence: "We therefore keep case B as our main estimate, since having separate data series for the two ocean layers provides information that influence the balance between negative and positive forcings, due to their different time evolution."

Small comments: p 3 model section: should this refer here already to using the Andronova and Schlesinger model? also, what are the 7 parameters? this would be very useful to hear what they parameterize and which parameter uncertainties are systematically investigated and which are not. this is probably retrievable from earlier papers but worth reiterating here.

We have included a paragraph describing the SCM model at the beginning of the model section and added a table in the supplementary material with the SCM parameters and their prior uncertainty ranges.  See our response to similar comments from reviewer 1.

Figure 5 discussion and caption: It would be helpful for a reader to understand what case B,C,D are from the caption - I got it on second but not first reading. (eg averaging rather than separate treatment etc).

The text in appendix D describing the test in case C and D is now merged with the main text. We hope this will make it clearer. In addition, we have added text to the caption. The new caption for figure 5 is now:

"Figure 5: Observed and fitted (posterior mean) total OHC using several OHC dataset (case B: separate OHC data above and below 700m and C: merge OHC data above and below 700m, left panel) and using only one dataset for the total OHC (case D, right panel). The shaded areas indicate the 90% CI."

In figure 4, I find it slightly confusing that one of the OHC series is systematically outside the estimated 90% range. can this be explained in the text?

See our response to a similar comment from reviewer 1.

**Short comments by N. Lewis**

In my view the article, while in principle suitable for publication by ESD, would be much improved if the following issues were addressed:

1. The information provided on the results is too limited. The median is a more appropriate best estimate measure than the mean for skewed distributions such as that for ECS. That is why the IPCC AR5 report gave medians, but not means, for all the observationally-based ECS estimates that it showed (Figure 10.20b). The medians should be shown, at least for the ECS and TCR posteriors, either instead of or in addition to the means, and likewise given in the Abstract and the main text. It is also slightly strange, for a Bayesian analysis, that the posterior PDF for ECS is only shown in panel j) of Figure S1.

As a response to this point, we have added the medians as triangles in Figure 1. Figure 1 will now also include a panel for TCR, and the medians are included in that figure as well. The median numbers are also included in the abstract and in the main text in addition to the mean values for case A-E.

The posterior PDFs for $ECS_{inf}$ and TCR for case A to E are added to the supplementary material as Figure S2.

2. The TCR estimate is of interest to readers as well as the ECS estimate, but it only seems to feature in Figure S2, with no values given. The Main analysis median TCR estimate and its 5-95% uncertainty range should be stated in the main text and, preferably, also in the Abstract.

As stated above, a panel for the TCR values are now included in Figure 1. We have included the main analysis median and 5-95% range in the main text and added a sentence regarding TCR in the abstract.

3. The study uses a subjective Bayesian analysis. The priors used likely have a major influence on the results, but finding out what they are requires referring to both Skeie et al 2014 and Aldrin et al 2012. A wide uniform prior seems to be used for ECS. It is well known that doing so biases ECS estimation upwards and greatly fattens the upper tail of the posterior. (Annan and Hargreaves 2011: "We show that the popular choice of a uniform prior has unacceptable properties and cannot be reasonably considered to generate meaningful and usable results."; Lewis 2014). Using a noninformative joint prior would produce estimates that were at least approximately unbiased, but calculating one could be difficult. Providing results based purely on the joint likelihood function, using a frequentist profile likelihood method, would be a reasonable alternative. If, as seems likely to be the case, the profile likelihood peaks at approximately the same point as the marginal likelihood for ECS (being the mode of the posterior, as a uniform prior for ECS is used) then the maximum likelihood estimate for ECS would be _1.75 K.

Also, showing what the characteristics of the ECS posterior are when a prior for ECS that is uniform in 1/ECS (and therefore is proportional to 1/ECS^2) is used would be helpful. That prior will be close to noninformative. [Given that fractional uncertainty in forcing (RF) is approximately symmetrical (Fig. 3(b)) and dominates that in GMST (and in ocean heat uptake), a uniform prior will be approximately noninformative for 1/ECS, and on a change of variable to ECS a uniform prior becomes 1/ECS^2.]

To demonstrate the sensitivity of the prior for ECS, we have computed an alternative estimate of ECS where the prior for 1/ECS is uniformly distributed, see our answer to a comment from reviewer 2.

4. The stepwise update results are interesting, but difficult to interpret in the absence of adequate quantitative information as to the changes in data values and uncertainty ranges involved.

We have now merged the text in the appendix B and the text on page 5-6 for the stepwise update to make this clearer, and we have added the 90% CI for each step in the text.

---

## Author Comment (AC3) · 23 Mar 2018

The comment was uploaded in the form of a supplement:
https://www.earth-syst-dynam-discuss.net/esd-2017-119/esd-2017-119-AC3-supplement.pdf